# Mixed recurrent connectivity in primate prefrontal cortex

**Evangelos Sigalas, Camilo Libedinsky** *

National University of Singapore, Singapore, Singapore

* camilo@nus.edu.sg

## Abstract

The functional properties of a network depend on its connectivity, which includes the strength of its inputs and the strength of the connections between its units, or recurrent connectivity. Because we lack a detailed description of the recurrent connectivity in the lateral prefrontal cortex of primates, we developed an indirect method to estimate it. This method leverages the elevated noise correlation of mutually-connected units. To estimate the connectivity of prefrontal regions, we trained recurrent neural network models with varying percentages of bump attractor connectivity and noise levels to match the noise correlation properties observed in two specific prefrontal regions: the dorsolateral prefrontal cortex and the frontal eye field. We found that models initialized with approximately 20% and 7.5% bump attractor connectivity closely matched the noise correlation properties of the frontal eye field and dorsolateral prefrontal cortex, respectively. These findings suggest that the different percentages of bump attractor connectivity may reflect distinct functional roles of these brain regions. Specifically, lower percentages of bump attractor units, associated with higher-dimensional representations, likely support more abstract neural representations in more anterior regions.

**Data availability statement:** All data and code used for running experiments, model

## Author summary

The strength of the connectivity between neurons is a fundamental property of brains that allows them to store memories and perform computations. This connectivity strength can be measured by recording the intracellular voltage changes evoked by presynaptic activation. However, this is technically unfeasible in large neural networks. Alternatively, this connectivity can be estimated using electron microscopy. However, these ultrastructural anatomical maps are time consuming, and not currently available in mammals. Thus, here we developed a method to estimate the connectivity of a network using extracellular physiological measurements. We measured pairwise correlations between neurons in the prefrontal cortex of monkeys performing a cognitive task, and then compared these values with multiple artificial neural network models trained to perform the same task as the monkeys, but initialized with different proportions of bump-attractor connectivity. Using this method, we estimate that approximately 20% of frontal eye field and 7.5% of dorsolateral prefrontal cortex neurons have a bump-attractor-like connectivity. We interpret these findings in the context of the functional roles of these regions in cognitive operations.

fitting, and plotting is available on a GitHub repository at https://github.com/esigalas/MixedConnectivityModels

**Funding:** Ministry of Education of Singapore (https://www.moe.gov.sg/) grants MOE-T2EP30121-0010 (C.L.) and MOE2017-T3-1-002 (C.L.) The funders had no role in study design, data collection and analysis, decision to publish, or preparation of the manuscript.

**Competing interests:** The authors have declared that no competing interests exist

## Introduction

Working memory, the ability to maintain and manipulate information without external input, is a fundamental cognitive function. Single neurons in the prefrontal cortex of primates and the hemodynamic response in the prefrontal cortex of humans selectively increase during memory maintenance [1–3]. Furthermore, lesioning [4], inactivating [5], and microstimulating [6] the prefrontal regions interferes with working memory maintenance. Thus, the prefrontal cortex plays an important role in maintaining working memories.

Recurrent neural networks (RNN) are networks that contain units that can mutually influence each other (i.e., unit A may influence – directly or indirectly - the activity of unit B, while unit B may also influence the activity of unit A). RNN models with attractor states have been shown to model many neural processes in the brain [7]. Specialized versions of such computational models (with local excitation and long-range inhibition), referred to as *bump attractor models* (Compte et al., 2000 [8]; Fig 1, left), can replicate several properties of prefrontal networks, suggesting that the bump attractor connectivity is present in these regions [9,10]. While other model classes may also be present in the PFC, we are only aware of experimental evidence supporting the presence of bump-attractor networks in the PFC. Thus, in this study we focus on this model class.

The bump attractor connectivity may be a conspicuous property of prefrontal networks, such that most neurons have the connectivity (e.g., such an arrangement is observed in the ellipsoid body of the drosophila brain; Kim et al., 2017) [11]. Alternatively, prefrontal

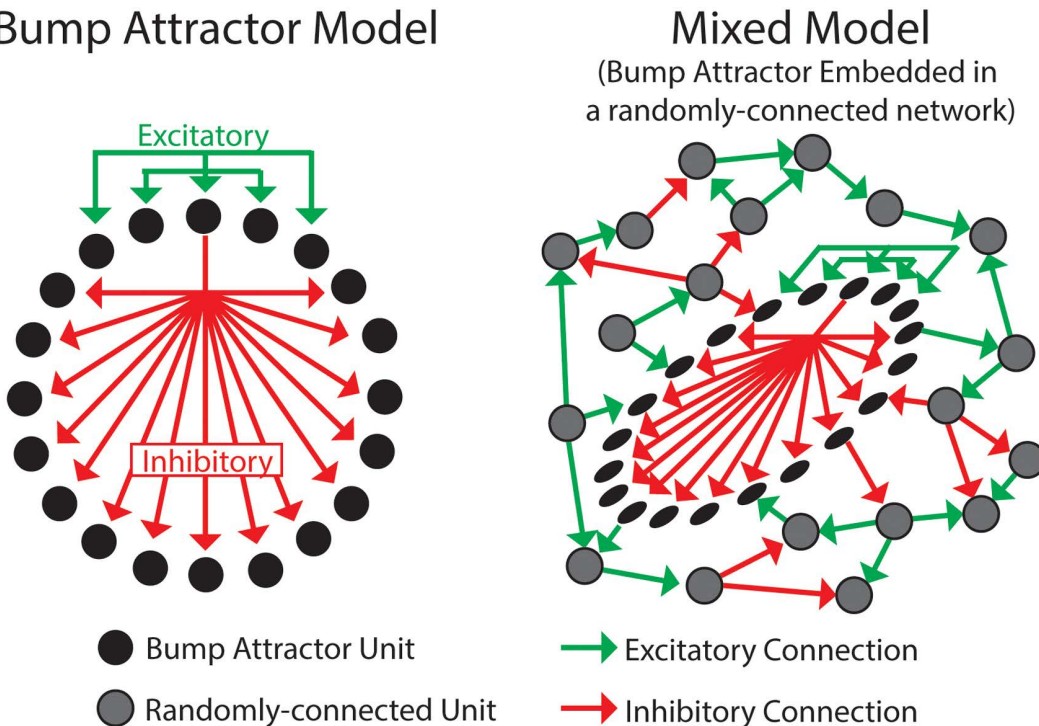

**Fig 1. Recurrent neural network models.** *Left*. The Bump attractor Model consists of excitatory connections (green) between units that receive similar inputs and inhibitory connections (red) with units that receive different inputs (inputs are not shown in the figure, but adjacent units receive similar inputs). Connections are only shown for a unit at the top, but the other units have similar connectivity. *Right*. The Mixed model has a fraction of units connected with the bump attractor connectivity, while the rest are randomly connected (note that these 2 sub-networks are mutually connected).

networks could contain *mixed connectivities* (Fig 1, right), where a subset of the neurons are part of a bump attractor network, while the rest of the neurons are connected differently, such as randomly-connected. To determine whether the prefrontal cortex primarily contains bump attractor connectivity or a mixed connectivity we could analyze a detailed anatomical map, as has been done before in the fly brain [12]. Unfortunately, this detailed anatomical information is unavailable for the primate brain. To sidestep this problem, we developed an indirect way to estimate the anatomical connectivity of the prefrontal cortex based on a physiological property of pairs of neurons: their noise correlation.

Noise correlation refers to the co-variability of the activity of pairs of neurons (while controlling for task-related changes in activity). Noise correlations offer valuable insights into the underlying mechanisms of information processing and coding in the brain, and they can provide important information about the functional connectivity of neurons in a network [13]. Due to the strong recurrent connectivity between bump attractor units, pairs of units with overlapping selectivity should have higher noise correlations than randomly-connected pairs with overlapping selectivity. As expected, we found that mixed RNN models trained to perform the same working memory task as the monkeys revealed higher noise correlations between bump attractor units than randomly connected ones. With this tool, we estimated the percentage of bump attractor neurons (within a mixed network) in 2 adjacent prefrontal regions: the frontal eye field (FEF) and the dorsolateral prefrontal cortex (DLPFC). We found that the noise correlation of the FEF and DLPFC was consistent with these regions having ~20% and ~7.5% of neurons with bump attractor connectivity, respectively. We discuss these differences in the context of the different functional roles of these 2 brain regions.

## Results

We trained 3 macaque monkeys (*macaca fascicularis*) to perform tasks that required the maintenance of working memory information during a delay period (Fig 2A). We measured the activity of neurons in two different prefrontal regions (areas 8a and 9/46, henceforth referred to as FEF and DLPFC) while the animals performed the tasks (Fig 2B). If two neurons are mutually exciting each other, they should have overlapping selectivity, and they should also show correlated noise fluctuations. We used these properties of bump attractor neurons to analyze the neural data in search of pairs of neurons with putative bump attractor connectivity.

In both prefrontal regions, we identified selectively active neurons during the delay period (FEF: 46%; DLPFC: 39%; Fig 3A and S1 Table). To calculate the noise correlation between pairs of selective neurons, we z-scored the activities for each location across trials to remove task-related activity from the analysis. For each neuron, we stitched together its z-scored activity during the first delay period (300 - 1300 ms after target onset) into a one-dimensional time series. This time series was used to measure the Pearson correlation coefficient between pairs of neurons. We then identified pairs of neurons with overlapping selectivity and calculated the proportion of these neuron pairs that showed a significant noise correlation (FEF: 36%; DLPFC: 10%; Fig 3B and S2 Table). We also counted the number of neurons that participated in these pairs since one neuron could participate in more than one pair (FEF: 44%; DLPFC: 29%; Fig 3B and S2 Table). Finally, we calculated the median Fano factor of neurons with overlapping selectivity (FEF: 1.14; DLPFC: 1.08; Fig 3C).

The bump attractor connectivity is a local property that reflects anatomical connections between neighboring neurons. As such, we would not expect to find a significant number of correlated neurons across brain regions. Thus, as a control, we conducted the same analyses on selective neurons across FEF and DLPFC (S1 Fig). This analysis showed a low number of correlated pairs (3/90, 3%), supporting the observation that these noise correlations are non-trivial and likely reflect the underlying connectivity.

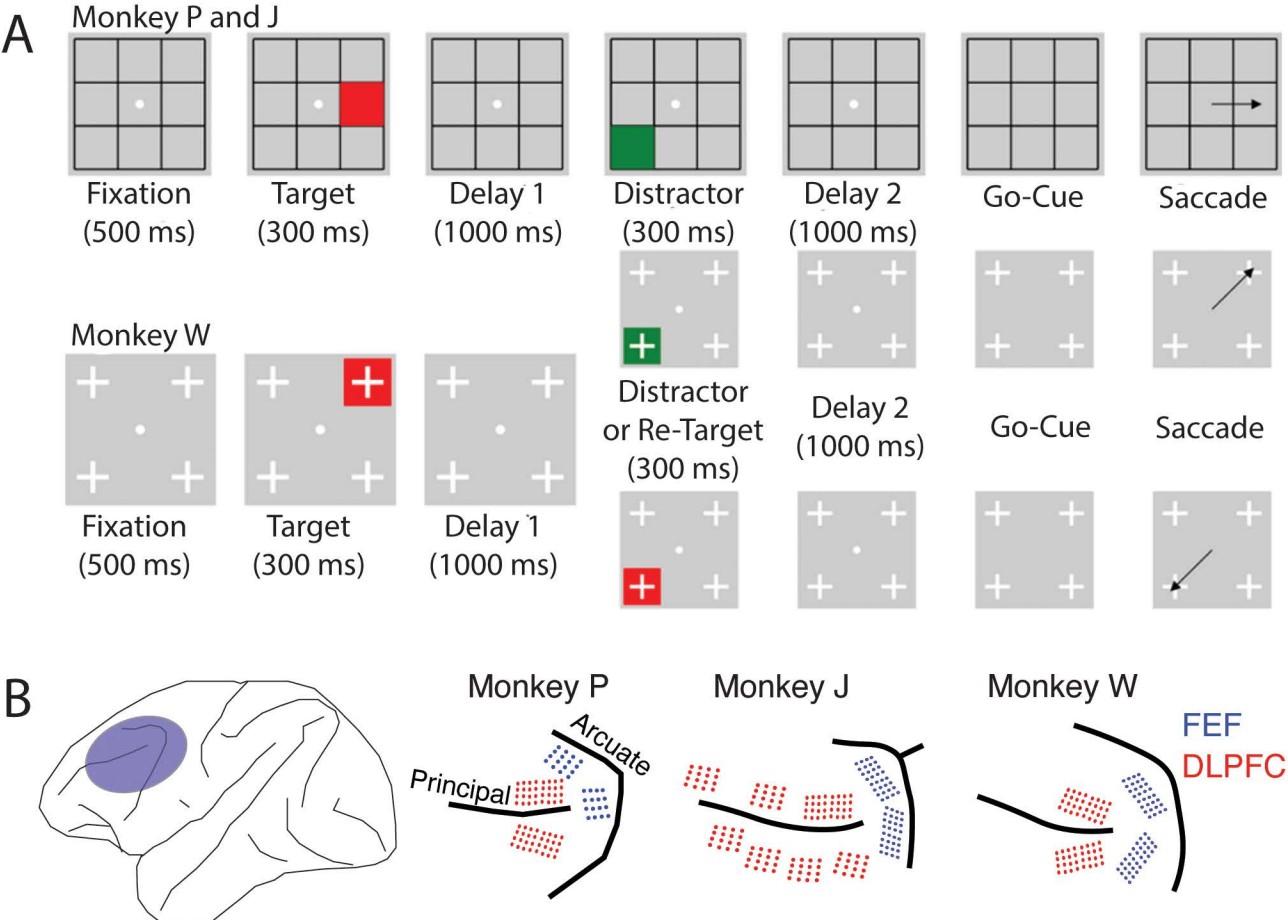

**Fig 2. Task Description and electrode locations** *(A) Task description. Both tasks are different versions of a visually-guided delayed saccade task. Trials were initiated by fixating for 500 ms on a central fixation spot, after which a red square (or target) was shown for 300 ms in 1 out of 8 locations (Monkeys P and J) or 1 out of 4 locations (Monkey W). After a 1000 ms Delay 1 period, a second stimulus was presented for 300 ms in a different location from the target. For Monkeys P and J, this second stimulus was always a green square, while for Monkey W, the second stimulus had a 50% chance of being a green square and a 50% chance of being a red square. The task rule for both monkeys was to report the location of the last red square seen. Thus, the green square, if shown, served as a distractor. After the second stimulus, a 1000 ms Delay 2 period was followed by a go-cue, which was the disappearance of the fixation spot. The monkeys had to saccade to the remembered location within 500 ms to receive a juice reward (B) Location of implanted electrode arrays. In the three monkeys, we chronically implanted electrode arrays in the pre-arcuate region, which includes the FEF (blue), and along the dorsal and ventral banks of the principal sulcus, denoted as DLPFC (red). For all arrays, electrodes along the sulcus were longer (5 – 5.5 mm), while further from the sulcus they were shorter (1 – 1.5 mm).*

To determine whether the results identified in the FEF and DLPFC are consistent with the bump attractor or the mixed model, we initialized mixed models with different percentages of bump attractor connectivity (5% to 100%, where 100% corresponds to a pure bump attractor connectivity), and then trained these RNNs to perform the same tasks as the monkeys (the training modified the recurrent weights of all the units in the model, including the recurrent weights initialized with the bump attractor connectivity). We searched for models that matched the physiological properties found in the neural data. In particular, we ensured that the models had a similar number of (1) selective neurons, (2) significantly correlated pairs of neurons, (3) neurons that participate in the correlated pairs, and (4) Fano factor (these properties are highlighted in the red boxes in Fig 3). To achieve this, we explored two meta-parameters: the proportion of units with bump attractor connectivity and the level of

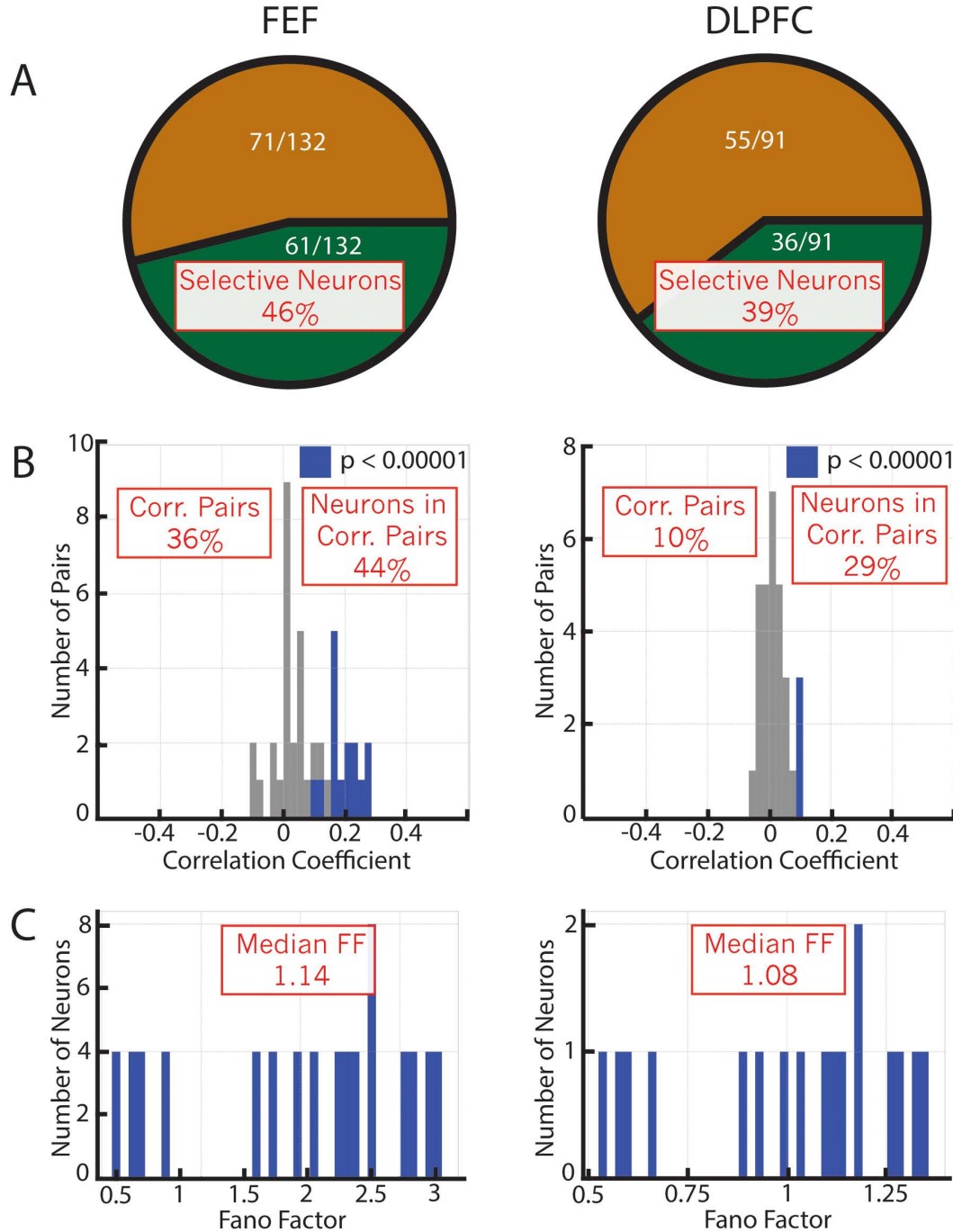

**Fig 3. Noise correlation analysis in PFC regions.** *(A) The proportion of neurons with selective activity increases for FEF (left) and DLPFC (right). (B) The correlation coefficient for the neuron pairs (blue bars: significant values; gray bars: non-significant values). (C) Fano factor for the neuron pairs. The red squares highlight the properties of this data that were used to select the RNN models in subsequent analyses.*

recurrent noise. Searching this 2-dimensional space, we found that a model with 20% bump attractor connectivity and a noise level of 0.08 matched the properties of the FEF neural data, while a model with 7.5% bump attractor connectivity and a noise level of 0.08 matched the properties of the DLPFC neural data (Fig 4). We confirmed that bump attractor units with

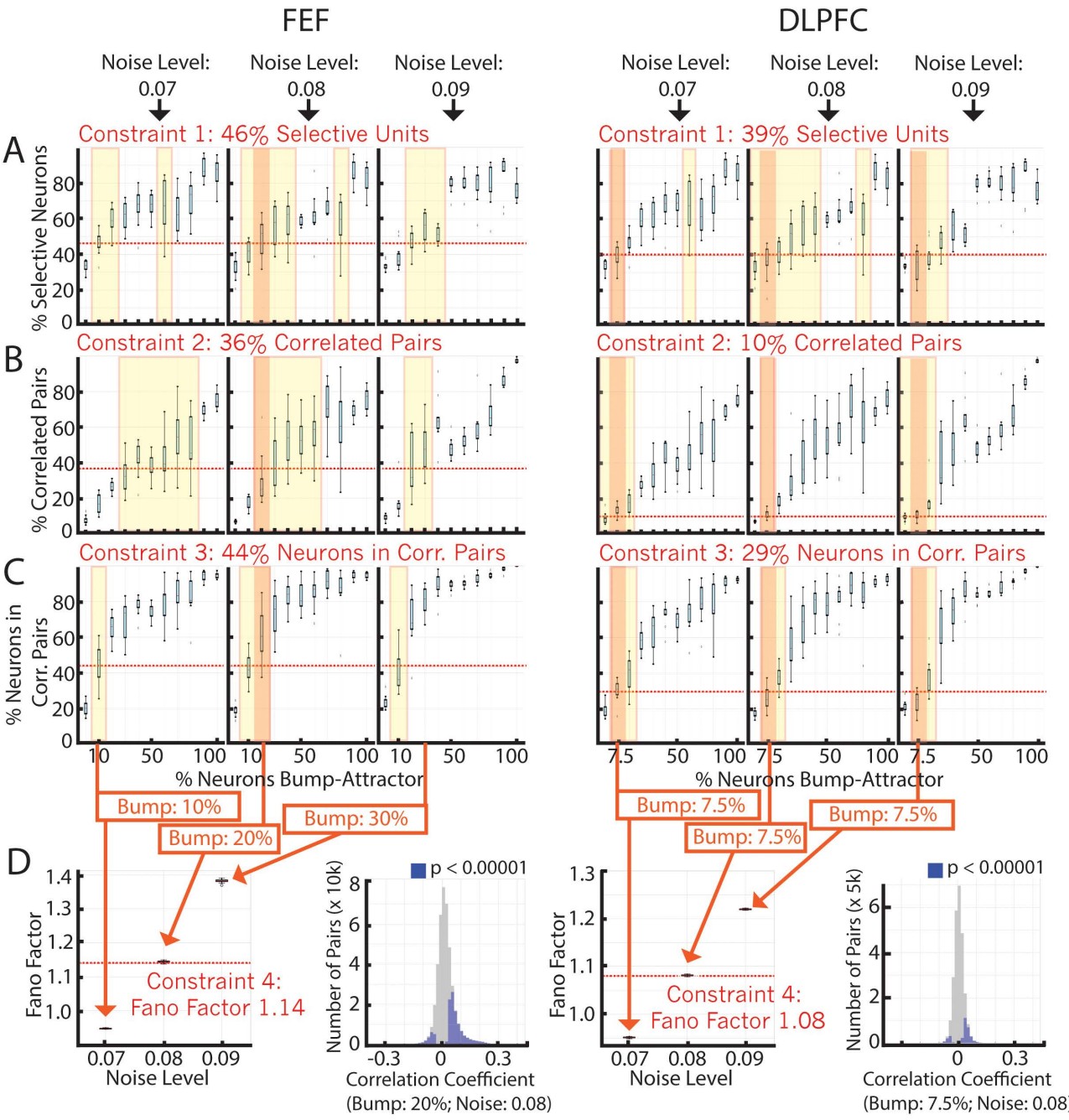

**Fig 4. Matching model properties to FEF (left) and DLPFC (right) constraints.** *(A) Percentage of selective neurons for models trained using different percentages of bump attractor units (x-axis) for different noise levels (left to right: 0.07, 0.08, 0.09). (B) Same as (A) but for the percentage of correlated pairs. (C) Same as (A) but for the percentage of neurons in correlated pairs. (D) (left) Fano factor for models that consistently match the properties in A-C: yellow window highlights models that match the specific property, and orange window highlights models that match all properties in A-C (for noise levels without a match for all properties, we selected the bump percentages that matched at least 2 of them). (Right) The correlation coefficient for the neuron pairs (blue bars: significant values; gray bars: non-significant values).*

overlapping selectivity had higher noise correlations than randomly-connected units; for the 20% model, bump neurons had a median r value of 0.13 (0.16 std.) while the median of randomly-connected neurons was 0.05 (0.13 std.) (p < 0.001), and for the 7.5% model bump

neurons had a median r value of 0.08 (0.11 std.) while the median of randomly-connected neurons was 0.01 (0.06 std.)(p < 0.001). We could not find models that matched the physiological properties of the data by training purely random models (0% bump attractor connectivity) (S2 Fig).

To determine whether these identified mixed models matched the population decoding properties of FEF and DLPFC, we performed cross-temporal decoding on the neural data and their corresponding models (Fig 5A). We quantified two decoding properties: (1) the amount of decodable information, which can be measured with a time-specific decoder, and (2) the stability of the code used to encode this information, which can be measured by how well a decoder trained at one time point generalizes to another time point [14]. In both regions

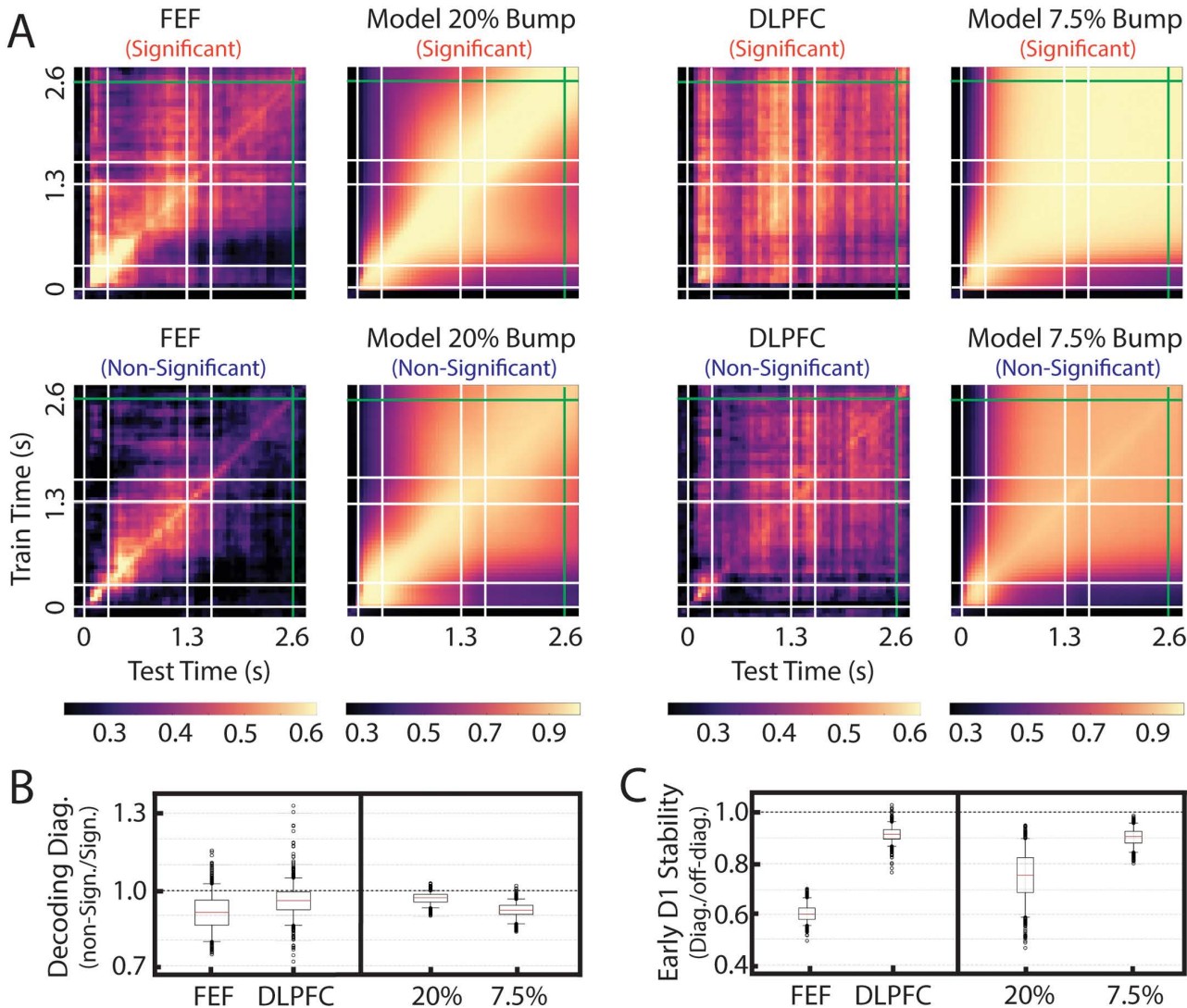

**Fig 5. Population decoding.** *(A) Cross-temporal decoding of FEF, its matching model (20% bump), DLPFC, and its matching model (7.5% bump) for neurons/units that participate in at least one significantly correlated pair (top plots) and those that do not (bottom plots). (B) Decoding accuracy of the diagonal (0 to 2.6 s) of non-significantly-correlated units divided by significantly-correlated units. (C) Code stability in the first 500 ms of Delay 1 (0.3 to 0.8 s) for significant neurons/units. Code stability is quantified as the mean time-specific decoder performance (0.3 to 0.8 s) divided by the mean performance of decoders trained during the 0.3 to 0.8 second period and tested between 0.8 and 2.6 s.*

and models, neurons/units forming part of one or more significantly correlated pairs (top plots in Fig 5A) had more decodable target information (p < 0.001; Fig 5B). Furthermore, the generalizability of the memory code during the first 500 ms of the Delay 1 period was higher in DLPFC and the 7.5% model compared to that found in FEF and the 20% model (p < 0.001; Fig 5C). One difference between the brain regions and their corresponding models is that the overall decoding performance in the models was higher than in the brain regions. We speculate that this is the result of an underestimation of the noise level in our models (see discussion). It is important to emphasize that these models (20% and 7.5%) were only selected based on single neuron and pairwise correlation data, so the observation that their population decoding behavior matches that of the data provides independent support for the proposed models.

## Discussion

There is evidence that the bump attractor connectivity is found in the lateral prefrontal networks (Wimmer et al., 2014) [10]. However, an outstanding question is how much of this connectivity is present in different prefrontal regions. Here, we show that a posterior prefrontal region, the FEF, has a higher percentage of bump attractor connectivity (~20%) than a more anterior prefrontal region, the DLPFC (~7.5%). These differences could have important consequences for the functions of these different regions, as discussed below.

The existence of bump attractor connectivity in the PFC has been criticized because certain properties of bump attractor networks are not shared by the PFC. For example, the PFC does not contain an orderly organization of neurons with the precise connectivity required by the bump attractor model [15,16]. However, a recent study showed that PFC neurons with similar spatial or shape selectivity are more likely than chance to be encountered at short distances from each other [17], providing a possible solution to this line of criticism. Furthermore, the bump-attractor connectivity does not need to have cell bodies localized in close proximity. Rather, all that is needed is for neurons, wherever they are localized, to receive structured inputs and connect to each other in a structured way (the bump-attractor connectivity).

An additional criticism is that the selectivity of PFC neurons is not always consistent between stimulus and delay periods, while that is the case for bump attractor networks [18,19]. Our results, which suggest that the bump-attractor connectivity is only present in a small percentage of neurons, could reconcile the observation that some PFC neurons maintain the selectivity between stimulus and delay, but the majority do not [18,19]. We should emphasize that we are not claiming that the canonical bump-attractor connectivity is present in the PFC, since our initialized bump-attractor connectivity is modified after training. What we are claiming is that a connectivity resembling the bump-attractor (with units with similar selectivity exciting each other and inhibiting units with dissimilar connectivity) is present in different proportions in FEF and DLPFC.

One of the most clear differences between our models and the PFC data is that the decoding of working memory information in the models generally exceeded that of the brain data (Fig 5). We speculate that this is the result of an underestimation of the noise level in the models. We explored different levels of noise in the models, and settled on a level that allowed us to match all the constraints derived from the data, including the fano factor. However, the fano factor of rate networks, like the ones we employed for our models, may not correspond perfectly to the fano factor of spiking networks, like the ones recorded from the PFC regions, because we are not considering the variability induced by the stochasticity of spike generation [20]. Thus, we speculate that a more accurate estimate of PFC connectivity could be achieved with spiking networks instead of rate networks.

The lateral prefrontal cortex has been parcellated into separate brain regions, including the FEF and DLPFC, based on anatomical and functional properties [21–23]. These regions appear to be organized along an anterior-posterior global functional gradient [22,24–28] which may reflect a functional hierarchy, with posterior regions tracking changes in the organism and environment while anterior regions support abstract neural representations and complex action rules [27,29]. Anterior and posterior prefrontal regions differ in inter-regional connectivity patterns [23], which likely contributes to the functional differences between regions. Here, we describe an additional factor that may contribute to these functional differences: different proportions of bump attractor connectivity.

State-space analyses of neural populations have revealed population-level mechanisms involved in representing information and performing computations over these representations [30,31]. An important feature of recurrent neural network dynamics is the dimensionality of the network's representations: the number of principal components required to account for a fixed proportion of variance in the data [32]. High-dimensional neural representations enable flexibility in processing, while low-dimensional neural representations enable stable and robust representations [9,33–35]. The dimensionality of a network depends on its inputs and recurrent connectivity [36,37]. Since the bump attractor connectivity is low-dimensional, our observation that an anterior prefrontal region (DLPFC) contains a lower proportion of bump attractor connectivity implies that this region contains higher-dimensional neural representations. This observation is consistent with the higher proportion of neurons with non-linear mixed selectivity observed in the DLPFC compared to the FEF [38] since non-linear mixed selectivity supports higher dimensional representations [39]. The higher dimensionality of the DLPFC is consistent with its role in more abstract and complex neural representations.

It is important highlight potential issues with the analysis and the model selection. Firstly, we assume that the bump-attractor connectivity is present in the PFC, which is only one network class out of the many possible ones. While this assumption is supported by indirect empirical evidence, it is plausible that it is wrong, and a different class of networks could lead to a better match with the data. It is also possible that non-stationarities in our data affected our cross-correlation estimation [40]. We minimized non-stationarities by using neural data during the delay period, which does not contain any changing stimuli nor changing cognitive demands. Furthermore, this activity was z-scored per location. The possibility remains that some global sources of non-stationarity may exist (such as fluctuations of alertness), which would affect our results. However, for this to be the case, we would need these global sources of non-stationarity to affect single neurons in each area differently, and also with different temporal profiles in FEF and LPFC, since we found very few neurons correlated across regions, which seems unlikely. Regarding the models, they contain several simplifications and assumptions that may influence the precise percentages of bump attractor connectivity that we estimated. First, the non-bump attractor network was modeled as a randomly connected network. However, randomness in the connections is not a necessary feature since other connectivities may coexist with the bump attractor connectivity. Second, we did not include short-term plasticity in the model units, which may be relevant for the encoding of working memory [41–44]. Third, FEF and DLPFC are interconnected networks, but we modeled them separately. Multi-region models may allow better estimates of the proportion of bump attractors in these networks [45]. Fourth, we modeled each region as having 1 attractor network, while it is possible that each region has multiple attractor sub-networks [46]. Thus, the precise values proposed (20% in FEF and 7.5% in DLPFC) should be taken as initial estimates of these regions' recurrent connectivity.

## Methods

### Data pre-processing

We recorded 132 neurons in the FEF and 91 neurons in the DLPFC of 3 monkeys while performing a delayed saccade task. We removed the neurons with low spike rate (< 10 Hz) during the Delay 1 period (300 - 1300 ms after target onset) for all locations. To determine the selectivity of each neuron, we performed a one-way ANOVA on the mean activity during the second half of the Delay 1 period (800 - 1300 ms after target onset) across locations. Among selective neurons, to determine which target locations had elevated activity during the delay period (i.e., significant locations), we performed a post-hoc one-tailed t-test between the mean neural activities of all target locations and the location with the lowest mean neural activity. Finally, we selected those locations as selective only if their mean activity was higher than during the baseline period (100 ms before stimulus onset; Fig 6).

### Noise correlation analysis

For the noise correlation analysis, we used the neural activity during the Delay 1 period (300 - 1300 ms after target onset). We wanted to compare how neurons vary around their mean activity on a trial-by-trial basis. To normalize the firing rate changes of each neuron, we z-scored their activities per location. These z-scored activities of each neuron were used to measure their noise correlations using the Pearson correlation coefficient index (PCI) formula.

For each session, we identified the selective locations of each cell as described above. Then, for pairs of neurons with overlapping selectivity, we concatenated their z-scored activity across trials for these locations. This resulted in two time series, one for each cell, which we used to calculate the PCI between the two cells. Finally, we combined the calculated PCIs from all the recording sessions of all monkeys in two histograms, one for each of the two brain regions (Fig 3B). An equivalent method was used to generate the histograms of the models shown in Fig 4D.

### Mixed connectivity model

We tested mixed connectivity models that were initialized with different proportions of two types of connectivity:

- A group of units connected with the bump attractor connectivity. These units excite other units that receive similar inputs while inhibiting units that receive different inputs. This type of connectivity has been inferred in the PFC of macaque monkeys [10].

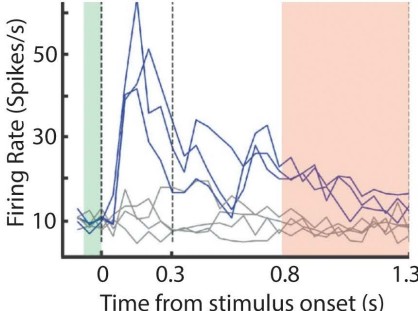

1. Mean activity in D1 period (0.3-1.3) > 10 Hz

2. ANOVA across locations in second half of D1 period (orange)

3. A location is significant (blue traces) if:
 a. Activity is significantly larger than the location with the lowest activity, and
 b. Activity is larger than baseline activity (green)

**Fig 6. *An illustration of the method used to determine the significant locations in individual neurons/units.*** *The same method was applied to neurons in the brain and units in the models.*

- A group of units connected randomly. Wherever in the manuscript we mention randomly initialized matrices we mean with the use of the Xavier initializer (Glorot and Bengio, 2010).

These two groups of units were randomly interconnected. We refer to them as bump units and non-bump units, respectively. More specifically, we model the mixed connectivity of bump and non-bump units as a Recurrent Neural Network (RNN), which is governed by the following equations:

$$\tau dx^{(bump)} = \left(-x^{(bump)} + W_{rec}^{(bump)} \, r^{(bump)} + b_{rec}^{(bump)} + W_{in}^{(bump)} \, u\right)dt + \sigma_{rec}^{(bump)}\sqrt{2\tau d\xi}, \qquad (1)$$

$$r^{(bump)} = f\left(x^{(bump)}\right), \qquad (2)$$

$$\tau dx^{(non-bump)} = \left(-x^{(non-bump)} + W_{rec}^{(non-bump)} \, r^{(non-bump)} + b_{rec}^{(non-bump)} + W_{in}^{(non-bump)} \, u\right)dt + \sigma_{rec}^{(non-bump)}\sqrt{2\tau d\xi}, \qquad (3)$$

$$r^{(non-bump)} = f\left(x^{(non-bump)}\right), \qquad (4)$$

$$z = W_{out} r + b_{out} \qquad (5)$$

where u, x, and z are the input, recurrent state, and output vectors. $W_{in}$, $W_{rec}$, and $W_{out}$ are the input, recurrent, and output synaptic weight matrices. The input u includes the stimulus as well as the responses of the other group of units. $b_{rec}$ and $b_{out}$ are constant biases. dt is the simulation time-step, and $\tau$ is the intrinsic time scale of the recurrent units. $\sigma_{rec}$ is a constant to scale recurrent unit noise, and $d\xi$ is a Gaussian noise process with mean 0 and std 1. f is a non-linear activation function adapted from [10].

$$f(x) = \begin{cases} 0, & x < 0 \\ x^2, & 0 < x < 1 \\ \sqrt{4x-3}, & x > 1 \end{cases} \qquad (6)$$

The matrix, $W_{rec}^{(bump)}$ and $W_{rec}^{(non-bump)}$ have different structures. The bump recurrent weight matrix (Fig 7) has a diagonal shape with positive values near the diagonal and negative values elsewhere, such that a few units are connected via excitatory weights to each other while being connected to inhibitory weights with the rest of the network. In this way, the adjacent bump units can generate a self-sustaining bump activity when they receive a structured input signal [10]. More specifically, the synaptic weights of each bump unit were given by the following equation adapted from [10]:

$$\beta = e^{\kappa \cdot \cos(\theta)}, \qquad (7)$$

$$\gamma = e^{0.2\kappa \cdot \cos(\theta)}, \qquad (8)$$

$$W_{rec}^{(bump)} = \frac{\beta}{\sum \beta} - \frac{\gamma}{\sum \gamma}, \qquad (9)$$

Where κ=3 and θ∈[0, 2π]. The recurrent weight matrix of non-bump units does not have such a structure (Fig 7), and all the weights are randomly initialized. We assume that the inter-connectivity between the modules (bump to non-bump units) of the network is at 5%. This means that only 5% of the connections between the bump and non-bump units are randomly. The rest of the connections are set to zero. An example of the interconnectivity between the bump and non-bump units is shown in Fig 7.

The matrix $W_{in}^{(bump)}$ has a spatial structure (Fig 7), so each target input stimulus is mapped to a group of ten adjacent bump units. There is also an overlap between these groups of size one at both ends of the group. The matrix $W_{in}^{(non-bump)}$ has no spatial structure and is randomly initialized. The variable r in equation (5) is the concatenation of $r^{(non-bump)}$ and $r^{(bump)}$ and represents the responses of all the units in the network. Finally, the output z of the network is a 4-D array that represents the four possible locations of the target. The highest mean activity in this 4-D array during decision time is interpreted as the decision the network makes.

## Model parameters

We aimed to build models that matched the noise correlation analysis results from the monkey recordings. Since our models contain many parameters, we focused on the ones that presumably affect the noise correlation. These are the percentage of bump units in the model and each unit's independent recurrent noise $\sigma_{rec}$. The number of bump units in each model is set to 36. To change the percentage of bump units in each model, we select a different size of the total units in the network. The intrinsic time scale is selected as τ = 200 ms, and the time-step dt = 50 ms for all the trained models. Network parameters were chosen based on prior studies [20,47,48].

## Training parameters

All the mixed connectivity models were trained to perform the Target-Distractor Task. The loss function used was the mean square error between the z output of the model during decision-making and the ground truth output of each trial. The Adam optimizer was used with a learning rate of 0.001. For training, we used mini-batches of size 20. All the models

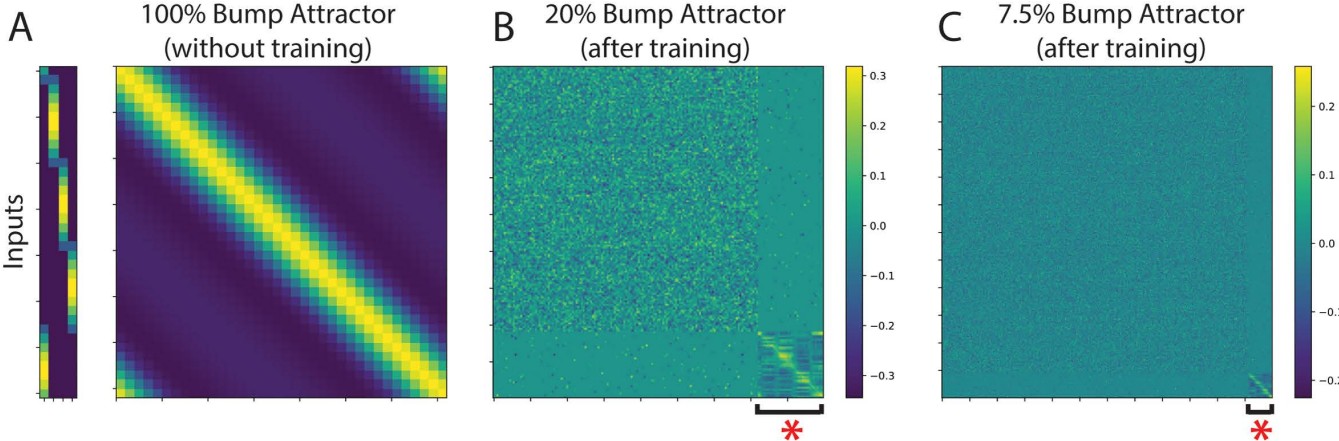

**Fig 7. Connectivity matrices.** *(A) The connectivity matrix of the bump attractor connectivity showing the input weights structure (left) and bump connectivity (right) where adjacent units excite each other (yellow color) and further away units inhibit each other (blue color). (B) Connectivity matrix of a mixed network with 20% bump attractor initialization (after training). The red asterisk shows the initialized units with the bump attractor connectivity. (C) Same as B but for the 7.5% bump attractor network.*

were trained until they achieved ~70% accuracy to match the monkeys' performance in the experiments. Finally, during training, we froze all the input weights of the network ($W_{in}^{(bump)}$, $W_{in}^{(non-bump)}$) and trained only the rest of the weights ($W_{rec}^{(bump)}$, $W_{rec}^{(non-bump)}$, $W_{out}$). The creation and training of the models was performed with the PsychRNN python framework. [49].

## Cross-temporal decoding

A decoder based on linear discriminant analysis (LDA) and principal component analysis (PCA) was built using the sklearn module in Python. We pooled the activity across recording sessions to create a pseudo-population of 36 neurons. We constructed 400 pseudo trials (100 for each location) and used ⅔ as the training set and ⅓ as the testing set. For each pseudo-population, we use the averaged neural activity of the second part of Delay 1 and Delay 2 of the train set to fit a PCA decoder. We reconstructed the train set data with the top $n$ principal components that explained at least 90% of the variance. Then, for each time point, we used the same PCA decoder to transform the test set data to the $n$ principal components. Finally, these two sets of train and test principal components were fed as input to an LDA decoder to determine the final decoding accuracy. This training and testing process was repeated for 200 pseudo-populations, and for each pseudo-population we repeated the split of the full dataset to train and test sets 5 times. This resulted in 1000 decoding matrices used for the results we reported in Fig 5.

## Behavioral tasks

For each trial, the animals maintained fixation for 3.1 seconds until a go-cue was given (the go-cue was the disappearance of the fixation spot). The trial was as follows:

$$Fixation\ (500\ ms) \rightarrow Stim.\ 1\ (300\ ms) \rightarrow Delay\ 1\ (1\ s) \rightarrow Stim.\ 2\ (300\ ms) \rightarrow Delay\ 2\ (1s)$$

Stimulus 1 was always a target (red square) that was presented at one of the eight locations in a 3 x 3 grid *(for Monkeys P and J)* or one of four locations at the corners of the same grid *(for Monkey W)*. For monkeys P and J, Stimulus 2 was always a distractor (green square), presented at a random location different from where the target was presented. For monkey W, Stimulus 2 had a 50% chance of being a green square (distractor) and a 50% chance of being a red square (re-target). If a re-target was presented, the animal was required to report the location of the Stimulus 2 target. All analyses were carried out on target-distractor trials for this task (i.e., we excluded the target/re-target trials). After Delay 2, a go cue (the disappearance of the fixation spot) signaled to the animal that he had to make a saccade toward the last red square presented. Saccades to the target location within a latency of 150 ms and continued fixation at the saccade location for 200 ms was considered a correct trial. An illustration of the tasks is shown in Fig 2A. For the cross-temporal decoding analysis of the task with 8 target locations, we only used the 4 locations in the corner to match those in the second version of the saccade task. The target-distractor version of the task was used to train all the mixed connectivity network models. The input to each network was a 4-D array, where each dimension represented the 4 locations of the grid. A noise of level $\sigma = 0.01$ was added to the input.

## Statistics

The Pearson Correlation Coefficient (PCC) in all cases was calculated using the Python scipy module as follows:

$$r = \frac{\sum (x - m_x)(y - m_y)}{\sqrt{\sum (x - m_x)^2 \sum (y - m_y)^2}}$$

where $m_x$ is the mean of the vector x, and $m_y$ is the mean of the vector y. The computed p-value is a two-tailed p-value. For a given sample with correlation coefficient r, the p-value is the probability that |r'| of a random sample x' and y' drawn from the population with zero correlation would be greater than or equal to |r|. The p-values of the Pearson correlation coefficients shown in Fig 3B are Bonferroni corrected.

To determine if the PCCs of the bump units in the model were significantly higher than those of the non-bump units, we ran a trimmed (Yuen's) t-test between the two distributions with a trim parameter equal to 0.49. The same trimmed (Yuen's) t-test was performed on all four distributions of the bar plots in Fig 5B to determine whether they were significantly lower than 1. Finally, for the distributions in Fig 5C, we applied the same t-test to the FEF-DLPFC distributions and the 20% - 7.5% model distributions.

## Supporting information

**S1 Table. Number of neurons recorded in each monkey and each region.** In parenthesis, the number of selective neurons is shown.
(DOCX)

**S2 Table. Number of neuron pairs with significant noise correlation and numbers of neurons in these pairs in FEF and DLPFC (note that one neuron can belong to multiple pairs).** In parenthesis, the number of significant pairs/neurons is shown.
(DOCX)

**S1 Fig. Cross-Region correlation analysis across regions.** The correlation coefficient for the neuron pairs with overlapping selectivity only for pairs that contain one neuron from FEF and one from DLPFC (blue bars: significant values; gray bars: non-significant values).
(TIF)

**S2 Fig. Purely random models (0% bump attractor architecture) failed to match the physiological constraints.** The middle column shows the parameters for random connectivity used in the rest of the manuscript (M: 0; STDV: 0.07). Left column shows results for connectivity with mean of -0.07 and STDV of 0.14, while the right column shows a network with mean 0 and STDV of 0.14. Networks with positive means did not learn the task.
(TIF)

## Acknowledgments

We thank Shih-Cheng Yen for comments on the analyses and Roger Herikstad for comments on the analyses and collecting the animal data.

## Author contributions

**Conceptualization:** Evangelos Sigalas, Camilo Libedinsky.

**Formal analysis:** Evangelos Sigalas, Camilo Libedinsky.

**Funding acquisition:** Camilo Libedinsky.

**Investigation:** Evangelos Sigalas, Camilo Libedinsky.

**Methodology:** Evangelos Sigalas, Camilo Libedinsky.

**Project administration:** Camilo Libedinsky.

**Software:** Evangelos Sigalas.

**Supervision:** Camilo Libedinsky.

**Writing – original draft:** Camilo Libedinsky.

**Writing – review & editing:** Evangelos Sigalas, Camilo Libedinsky.

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
