## [Decision Letter · Decision Letter 0]

3 Sep 2024

Dear Dr. Libedinsky,

Thank you very much for submitting your manuscript "Mixed recurrent connectivity architecture in primate prefrontal cortex" for consideration at PLOS Computational Biology.

As with all papers reviewed by the journal, your manuscript was reviewed by members of the editorial board and by several independent reviewers. In light of the reviews (below this email), we would like to invite the resubmission of a significantly-revised version that takes into account the reviewers' comments.

Synthesis

The manuscript was reviewed by two reviewers. Both found the paper very interesting. However, they have raised several concerns that need to be addressed in a major revision of the manuscript. Here are the main concerns:

The manuscript is quite sloppy and unclear in the description of the methods.

The locally of connectivity as required by the bumpNN is may not be consistent with the experimental data therefore it is important to discuss how BumpNN maps on the known cortical connectivity

Related to the previous point, some of the key controls necessary to establish BumpNN style connectivity are missing in the current version of the manuscript. The reviewers have made some suggestions in this regard.

Detailed comments from the reviewers are appended to help you revise and prepare a point by point reply. We look forward to reading the revised manuscript.

We cannot make any decision about publication until we have seen the revised manuscript and your response to the reviewers' comments. Your revised manuscript is also likely to be sent to reviewers for further evaluation.

Sincerely,

Arvind Kumar, Ph.D.

Academic Editor

PLOS Computational Biology

Daniele Marinazzo

Section Editor

PLOS Computational Biology

Synthesis

The manuscript was reviewed by two reviewers. Both found the paper very interesting. However, they have raised several concerns that need to be addressed in a major revision of the manuscript. Here are the main concerns:

The manuscript is quite sloppy and unclear in the description of the methods.

The locally of connectivity as required by the bumpNN is may not be consistent with the experimental data therefore it is important to discuss how BumpNN maps on the known cortical connectivity

Related to the previous point, some of the key controls necessary to establish BumpNN style connectivity are missing in the current version of the manuscript. The reviewers have made some suggestions in this regard.

Detailed comments from the reviewers are appended to help you revise and prepare a point by point reply. We look forward to reading the revised manuscript.

Reviewer's Responses to Questions

**Comments to the Authors:**

Reviewer #1: The authors attempt to test the potential existence of a bump attractor architecture in two regions of the primate prefrontal cortex (FEF and dlPFC) by matching several properties of electrophysiological activity in vivo with a recurrent neuronal network (RNN) trained to perform the same task as the monkeys. The network contains varying fractions of bump attractors (as compared to random connections without spatial structures) and is subject to varying degrees of external noise. These two parameters are varied to match the properties of interest. The authors find that activity in the FEF is consistent with an RNN with a higher percentage (20%) of bump attractors compared to dlPFC (7.5). They further compare decoding properties between models and data and find a number of similarities.

The author's approach to use computational models as a tool to uncover structures in experimental data is promising and timely. Also, the existence of attractor structures in the prefrontal cortex is a topic of intense debate currently, and this paper has the potential to contribute to this debate. However, in the current form, the analyses presented does not allow any valid statement about the existence of bump attractors in the prefrontal cortex. This is mainly due to the lack of proper controls: While properties such as increased noise correlations could indeed stem from a bump attractor synaptic architecture, there are many alternative explanations that cannot be excluded in the current research design. The simplest of these alternative explanations would be an increased level of overall connectivity (number of connections and/or synaptic weights) in the bump attractor architecture compared to the random connection pattern. In principle, evidence for the existence of bump attractors would require exhausting every possible (or at least, sensible) combination of parameters for random connectivity that may affect the correlation pattern, number of selective neurons, and so on. Only if neither of these combinations would yield a satisfactory fit to the data, one could conclude that a purely random connectivity pattern is insufficient to explain the observed data. Even in this case, a better fit with a bump attractor architecture would not exclude any other, non-random, pattern of connectivity that would explain the data as well (e.g. other models of working memory such as models based on short-term synaptic plasticity). But from such an analysis, one could at least conclude that the bump attractor architecture is superior in explaining the data over a random connection pattern - different from the study in the current form.

Apart from this main point, I noticed that many of the methodical details of the model are not well explained, and some choices may lead to problems in the analysis. In particular:

- According to the first equation in the methods section, there are no connections between the bumps and the random network, although this is implied in the text and Figure 1.

- The non-linear activation function needs to be specified.

- The details of the two network architectures are missing, although potentially crucial to the results. In particular, all equations and parameters need to be included in the Method section.

- Stitching parts of the time series together may cause problems when estimating correlations without further precautions, as it tends to induce non-stationarities, see Quiroga-Lombard et al., 2013, J Neurophysiol

- Key methodical concepts as RNNs and decoding properties need to be briefly explained in the Results section.

- The rationale behind the chosen decoding properties is unclear to me.

- The precise criteria for match between model and data for each of the properties under investigation are not mentioned. It seems that there is only a partial match for some of them.

- Figure 5A shows only little similarity between the decoding properties of the data and the corresponding model. This needs to be discussed.

As a final point, the discussion section seems very brief and somewhat superficial to me. I would have expected a discussion of the relation of the current research to similar studies in the past and a more thorough reflection on the implications of the findings and, in particular, the methodical limitations.

Reviewer #2: Review report for the manuscript entitled "Mixed recurrent

connectivity architecture in primate prefrontal cortex" by Sigalas and

Libedinsiky

In this manuscript the authors aim to estimate the degree of recurrent

"bump attractor network" architecture (BumpNN) in the DLPFC and FEF of

thee non-human primates while performing a visual delayed match to

sample working memory task. The method to do this relies on noise

correlation between selectively active neurons and network units

respectively. The network model used was a recurrent neural network

with a bump attractor ("ring") network embedded in a matrix of

randomly connected units, which also connected reciprocally to the

BumpNN network. By matchin the experimentally measured noise

correlation and other measures of activity between selectively active

PFC neurons and those of the BumpNN units with similar response

properties (and thus connected) they estimates that the degree of

BumpNN-ness was 7.5 and 20 %.

Major comments

Generally, I found this paper quite interesting, relevant and quite

well written, though somewhat sloppy and unclear when it comes to

e.g. detailed model descriptions. Some suggestions follow in the Minor

comments section.

I have one major concern though, and this relates to the non-critical

presentation of BumpNN as a "conspicuous" model of PFC. I think it can

have some relevance, but the way to map such an architecture to

neocortex is by no means obvious. A main criticism has been that the

local representation of the memory of an experimentally stimulated

position in the field of view predicted by the model, i.e. a local

connected and reverberating group of neurons goes against what many

experimental data and computational models of distributed memory

predicts. However, it is rather easy to get around this by emphasizing

not the geometrical locality but the mutual connectivity between the

units in the BumpNN. If the neurons in the cortical "BumpNN" is

instead distributed over a larger cortical area but with the same

connectivity, the dynamics and noise correlations would likely be

quite similar and the results of the paper still valid. For discussion

see e.g. Neural Mechanisms of Working Memory Accuracy Revealed by

Recurrent Neural Networks by Xie, Liu, Constantinidis and Zhou,

Frontiers 2022 (doi: 10.3389/fnsys.2022.760864).

What the architecture in the real neocortex actually looks like is

still not clear, so I request addition of this alternative view in a

prominent place in the manuscript, or alternatively a more detailed

explanation how to map the BumpNN to neocortical architecture. I would

recommend the term "bump attractor connectivity" or "bump attractor

properties" rather than "bump attractor architecture" at, for

instance, the first line in the Discussion section and several other

places. It will take some careful reformulations to make the text

consistent in this regard.

Minor comments

In the Methods section, the desciption of the RNN model with embedded

BumpNN needs to be expanded and improved. It lacks citations and

reference to standard RNN methodology. Is it etirely home cooked?

1. It should be clearly stated that no learning is used but the

BumpNN is entirely prewired.

2. What is "intrinsic time scale" of the unit. Does it refer to their

"membrane time constant" for instance. It seems too long (200 ms) for

this, so what does it correspond to?

3. Expressions like "near the diagonal", "mapped to a group of

adjacent bump units", "also some overlap between groups of adjacent

stimuli" need to be quantified to allow reproduction of the results.

4. The "- Training Parameters:" subsection came as a surprise since the

architecture is described as prewired. It should be clearly described

that this network also have an output population and how it is

connected and trained. This section should also be adjacent to the

other model subsections.

5. The "scipy" module should at least be the "Python scipy module".

6. The concept or low and high dimensionality of representations is

used in many places. But these concepts are not defined but should

be. An example definition is:

"Linear dimensionality: the number of principal components required to

account for a fixed proportion of variance in the data. For

low-dimensional data, the variance is concentrated in the first few

principal components with the largest eigenvalues. For

high-dimensional data, the variance is distributed across many

principal components with the eigenvalues of similar

magnitude. Intuitively, the dimensionality corresponds to the extent

of the linear subspace occupied by the data, or the number of separate

patterns exhibited by the data." [TA Engel an NA Steinmetz (2019), New

perspectives on dimensionality and variability from large-scale

cortical dynamics, Current Opinion in Neurobiology 58, 181-190.]

**Have the authors made all data and (if applicable) computational code underlying the findings in their manuscript fully available?**

Reviewer #1: Yes

Reviewer #2: **No: ** I did not find a Data Availability section in the manuscript.

PLOS authors have the option to publish the peer review history of their article (what does this mean? ). If published, this will include your full peer review and any attached files.

**Do you want your identity to be public for this peer review?** For information about this choice, including consent withdrawal, please see our Privacy Policy .

Reviewer #1: No

Reviewer #2: No
---

## [Decision Letter · Decision Letter 1]

20 Nov 2024

PCOMPBIOL-D-24-01220R1Mixed recurrent connectivity in primate prefrontal cortexPLOS Computational Biology Dear Dr. Libedinsky, Thank you for submitting your manuscript to PLOS Computational Biology. After careful consideration, we feel that it has merit but does not fully meet PLOS Computational Biology's publication criteria as it currently stands. Therefore, we invite you to submit a revised version of the manuscript that addresses the points raised during the review process. Please submit your revised manuscript within 60 days Jan 20 2025 11:59PM. If you will need more time than this to complete your revisions, please reply to this message or contact the journal office at ploscompbiol@plos.org. Please include the following items when submitting your revised manuscript: * A rebuttal letter that responds to each point raised by the editor and reviewer(s). You should upload this letter as a separate file labeled 'Response to Reviewers'. This file does not need to include responses to formatting updates and technical items listed in the 'Journal Requirements' section below.* A marked-up copy of your manuscript that highlights changes made to the original version. You should upload this as a separate file labeled 'Revised Manuscript with Track Changes'.* An unmarked version of your revised paper without tracked changes. You should upload this as a separate file labeled 'Manuscript'. If you would like to make changes to your financial disclosure, competing interests statement, or data availability statement, please make these updates within the submission form at the time of resubmission. Guidelines for resubmitting your figure files are available below the reviewer comments at the end of this letter. We look forward to receiving your revised manuscript. Kind regards, Arvind Kumar, Ph.D.Academic EditorPLOS Computational Biology Daniele MarinazzoSection EditorPLOS Computational Biology Feilim Mac GabhannEditor-in-ChiefPLOS Computational Biology Jason PapinEditor-in-ChiefPLOS Computational Biology **Additional Editor Comments:** As you will in the reviewers' comments, one of the reviewer is largely happy with the revision but most of the concerns raised by the second reviewer have remained unresolved. We therefore invite you one more time to adequtaly address the concerns of the reviewer. As I see it his/her comments are related to the robustness of the results and rigor in the approach -- we of course cannot compromise on that.  **Reviewers' comments:** Reviewer's Responses to Questions

**Comments to the Authors:**

Reviewer #1: The authors have improved the manuscript by providing a more detailed description of the model, a control regarding the overall connection strength and an extended discussion. While I appreciate these improvements, I also observe that my main critique has not been addressed: Their central claim ("the FEF, has a higher percentage of bump attractor connectivity (~20%) than […] the DLPFC", line 222-223) is still not supported by the data they present. What the authors show is that a model initialized with a higher percentage of bump attractor connectivity qualitatively reproduces spike train statistics from FEF better compared to a model with a lower percentage, which in turn better reproduces spike train statistics from DLPFC. No evidence for a causal relation between the models and the data has been shown. In the review of the first version, I suggested that (limited) evidence for such a relation could result from a rigorous set of controls that attempt to rule out any alternative explanation for the different model behavior except the percentage of bump attractor connectivity. The one single control the authors added is not remotely sufficient to tighten the evidence of causality. The fact that the models only loosely reflect the spike train statistics (as clarified in the author's responses) further weakens this evidence. The added methodical details on the model also allow for a more confident assessment that the necessary controls are next to impossible due to the large number of parameters and a lack of a clear theoretical foundation about the mechanisms underlying the model behavior. As a final remark, the authors claim in the beginning of the discussion that, to their knowledge, "this is the first study to attempt to infer network connectivity based on physiological properties" (line 225). By "physiological properties", I assume they mean the correlation and Fano factor of the spike trains. However, there is in fact a vast literature that attempted such inference. As a starting point, see de Abril, I. M., Yoshimoto, J., & Doya, K. (2018) for a review.

Given the lack of perspective to find a sensible set of controls, of theoretical foundation and of foundation in the literature, I regret that I have to recommend the rejection of the manuscript.

Reviewer #2: The formulation "... because certain properties of bump attractor networks are not shared by PFC neurons." compares network architecture with neuron properties, strictly speaking, which is unfortunate. Should change last part to "... shared by the PFC." or something equivalent.

The sentence "There is also some an overlap between ...", "an " should be removed.

**Have the authors made all data and (if applicable) computational code underlying the findings in their manuscript fully available?**

Reviewer #1: Yes

Reviewer #2: **No: ** No Data Availability Statement was found in the manuscript PDF file.

PLOS authors have the option to publish the peer review history of their article (what does this mean? ). If published, this will include your full peer review and any attached files.

**Do you want your identity to be public for this peer review?** For information about this choice, including consent withdrawal, please see our Privacy Policy .

Reviewer #1: No

Reviewer #2: No

 **Figure resubmission:**While revising your submission, please upload your figure files to the Preflight Analysis and Conversion Engine (PACE) digital diagnostic tool, https://pacev2.apexcovantage.com/. PACE helps ensure that figures meet PLOS requirements. To use PACE, you must first register as a user. Registration is free. Then, login and navigate to the UPLOAD tab, where you will find detailed instructions on how to use the tool. If you encounter any issues or have any questions when using PACE, please email PLOS at figures@plos.org. Please note that Supporting Information files do not need this step. If there are other versions of figure files still present in your submission file inventory at resubmission, please replace them with the PACE-processed versions. 
---

## [Decision Letter · Decision Letter 2]

27 Jan 2025

PCOMPBIOL-D-24-01220R2

Mixed recurrent connectivity in primate prefrontal cortex

PLOS Computational Biology

Dear Dr. Libedinsky,

Thank you for submitting your manuscript to PLOS Computational Biology. After careful consideration, we feel that it has merit but does not fully meet PLOS Computational Biology's publication criteria as it currently stands. Therefore, we invite you to submit a revised version of the manuscript that addresses the points raised during the review process.

Please submit your revised manuscript within 30 days Mar 29 2025 11:59PM. If you will need more time than this to complete your revisions, please reply to this message or contact the journal office at ploscompbiol@plos.org. Please include the following items when submitting your revised manuscript:

We look forward to receiving your revised manuscript.

Kind regards,

Arvind Kumar, Ph.D.

Academic Editor

PLOS Computational Biology

Daniele Marinazzo

Section Editor

PLOS Computational Biology

**Additional Editor Comments :**

Thanks for revising the manuscript. One of the reviewers is happy with the revision. However, concerns of one of the reviewers remain unaddressed. The reviewer makes a valid point. You essentially have explore only class of networks (i.e. the continuum of bum attractor type connectivity and random connectivity). Of course you will find that some combination of attractor type and random connectivity will be the best match for any given network. I do see value of the manuscript but the claims should be revised and it should be clearly stated that only one class of network were studied and another class of networks could have a better match. Therefore I invite you to revise the text once more and present the results in the right context.

**Journal Requirements:**

1) We have noticed that you have a list of Supporting Information legends for Supplementary tables in your manuscript. However, there are no corresponding files uploaded to the submission. Please upload them as separate files with the item type 'Supporting Information'.

2) Please provide a complete Data Availability statement in the online submission form.

**Reviewers' comments:**

Reviewer's Responses to Questions

Reviewer #1: I appreciate the clarification the authors made. I was aware that they did not aim to test for the existence of bump attractors in the cortex. However, this does not change my impression that the study suffers from a basic logical flaw: The authors show that a certain feature of a particular model, namely the degree of bump attractor connectivity, changes the quality of fit to data from two parts of the cortex. From that, they infer that the difference in bump attractor connectivity must also exist between the two parts of the cortex. This inference would only be valid if one could show that the change in bump attractor connectivity is not only sufficient, but also necessary to explain the difference in the fit to the two data sets. As already discussed, this is not the case: There are many other parameters that could affect the model, potentially in the same way as bump attractor connectivity. I offered one control which the authors indeed conducted, but I did not suggest that this single control was sufficient. Rather, I agree with the authors that investigating all thinkable controls would be infeasible. However, different from the authors, I conclude that the proposed inference is infeasible in general, at least while a more thorough theoretical framework is missing. If the authors could establish that the investigated activity signatures should be affected by bump attractor connectivity just in the proposed way, the (not too strong, by the way) empirical evidence could be seen as a confirmation of the theory. As it stands, the proposed difference between the two brain regions in bump attractor connectivity is merely a moderately supported hypothesis, which could be the starting point of a manuscript, but is, in my mind, insufficient for a full publication in PLoS Computational Biology. I regret that I cannot come to a more positive conclusion, while the final decision is of course with the editor.

Reviewer #2: I accept the revision done.

**Have the authors made all data and (if applicable) computational code underlying the findings in their manuscript fully available?**

Reviewer #1: Yes

Reviewer #2: None

PLOS authors have the option to publish the peer review history of their article (what does this mean? ). If published, this will include your full peer review and any attached files.

**Do you want your identity to be public for this peer review?** For information about this choice, including consent withdrawal, please see our Privacy Policy .

Reviewer #1: No

Reviewer #2: No

**Figure resubmission:**
---

## [Editor Report · Decision Letter 3]

11 Feb 2025

Dear Dr. Libedinsky,

We are pleased to inform you that your manuscript 'Mixed recurrent connectivity in primate prefrontal cortex' has been provisionally accepted for publication in PLOS Computational Biology.

Best regards,

Arvind Kumar, Ph.D.

Academic Editor

PLOS Computational Biology

Daniele Marinazzo

Section Editor

PLOS Computational Biology

We are now happy with the revision and rephrasing of the text about models that have been used to study the PFC activity. I am happy to recommend publication of the manuscript in its current form. Congratulations for a fine piece of work.

---

## [Editor Report · Acceptance letter]

PCOMPBIOL-D-24-01220R3

Mixed recurrent connectivity in primate prefrontal cortex

Dear Dr Libedinsky,

I am pleased to inform you that your manuscript has been formally accepted for publication in PLOS Computational Biology. Your manuscript is now with our production department and you will be notified of the publication date in due course.

With kind regards,

Anita Estes
